# A Modulation Layer to Increase Neural Network Robustness Against Data Quality Issues

**Mohamed Abdelhack**                    *mohamed.abdelhack.37a@kyoto-u.jp*
*Department of Anesthesiology, Washington University in St. Louis, St. Louis, MO, USA*
*& Krembil Centre for Neuroinformatics, Centre for Addiction and Mental Health, Toronto, ON, Canada*

**Jiaming Zhang**                    *jzhang99@usc.edu*
*Department of Computer Science, University of California, San Diego, CA, USA*

**Sandhya Tripathi**                    *sandhyat@wustl.edu*
**Bradley A Fritz**                    *bafritz@wustl.edu*
*Department of Anesthesiology, Washington University in St. Louis, St. Louis, MO, USA*

**Daniel Felsky**                    *daniel.felsky@camh.ca*
*Krembil Centre for Neuroinformatics, Toronto, ON, Canada*

**Michael S Avidan**                    *avidanm@wustl.edu*
*Department of Anesthesiology, Washington University in St. Louis, St. Louis, MO, USA*

**Yixin Chen**                    *ychen25@wustl.edu*
*Department of Computer Science and Engineering, Washington University in St. Louis, St. Louis, MO, USA*

**Christopher R King**                    *christopherking@wustl.edu*
*Department of Anesthesiology, Washington University in St. Louis, St. Louis, MO, USA*

**Reviewed on OpenReview:** *https://openreview.net/forum?id=MRLHN4MSmA*

## Abstract

Data missingness and quality are common problems in machine learning, especially for high-stake applications such as healthcare. Developers often train machine learning models on carefully curated datasets using only high-quality data; however, this reduces the utility of such models in production environments. We propose a novel neural network modification to mitigate the impacts of low-quality and missing data which involves replacing the fixed weights of a fully-connected layer with a function of additional input. This is inspired by neuromodulation in biological neural networks where the cortex can up- and down-regulate inputs based on their reliability and the presence of other data. In testing, with reliability scores as a modulating signal, models with modulating layers were found to be more robust against data quality degradation, including additional missingness. These models are superior to imputation as they save on training time by entirely skipping the imputation process and further allow the introduction of other data quality measures that imputation cannot handle. Our results suggest that explicitly accounting for reduced information quality with a modulating fully connected layer can enable the deployment of artificial intelligence systems in real-time applications.

## 1 Introduction

Despite the enormous academic and industrial interest in artificial intelligence, there is a large gap between model performance in laboratory settings and real-world deployments. Reports estimate that over 75% of data science and artificial intelligence projects do not make it into production (VentureBeat, 2019; Sagar, 2021; Chen and Asch, 2017). One difficult transition from the laboratory is handling noisy and missing data. Errors in predictor data and labels (Northcutt et al., 2021) at the training stage are well understood to produce poor pattern recognition with any strategy; garbage-in garbage-out. In the statistical learning literature, the effects of inaccurate and missing data on simple classifiers such as logistic regression is particularly well understood (Ameisen, 2020). As a result, datasets intended to train high-accuracy models are often carefully curated and reviewed for validity (Ameisen, 2020; Xiao et al., 2018). However, when faced with noisy data from a new source, these models may fail (L'Heureux et al., 2017). One special case is convolutional neural networks for machine vision; augmenting the dataset with partially obscured inputs has been shown to increase the network's ability to match low-level patterns and increases accuracy (Zhong et al., 2020).

These challenges are even more pronounced in applications that require high reliability and feature pervasive missing data at inference time, such as healthcare (Chen and Asch, 2017; Xiao et al., 2018). Electronic health records (EHR) can contain a high percentage of missing data both at random (keyboard entry errors, temporarily absent data due to incomplete charting) and informative or missing-not-at-random (MNAR) data (selective use of lab tests or invasive monitors based on observed or unobserved patient characteristics). Medical measurements also have non-uniform noise; for instance, invasive blood pressure measurement is more accurate than non-invasive blood pressure (Kallioinen et al., 2017). Another example is the medical equipment by different manufacturers that have various margins of error which affects the accuracy and hence the reliability of the measurement (Patel et al., 2007).

Mammalian brains have a distinct strategy to integrate multi-modal data to generate a model of the surrounding environment. They modify the impact of each input based on the presence and reliability of other signals. This effect can be observed dynamically in response to temporary changes in available inputs (Shine et al., 2019), as well as long-term as a compensation mechanism for permanent changes such as neural injuries (Hylin et al., 2017). For example, the human brain gives less weight to visual input in a dark environment and relies on prior knowledge and other sensory cues more. Unlike simply down-weighting low-accuracy data, replacement data with related information is up-weighted. This is usually modelled as a Bayesian inference process (Cao et al., 2019; Ernst and Bülthoff, 2004; Alais and Burr, 2004; Heeger, 2017). This modulation of different inputs is also observed in other organisms where the neural behavior of a neuron or a group of neurons can be altered using neuromodulators (Harris-Warrick and Marder, 1991; Abdelhack, 2022). Models of neuronal activity have also shown that modulation can play an important role in learning (Swinehart and Abbott, 2005), locomotion control (Stroud et al., 2018), context modulation (Podlaski et al., 2020), and reinforcement learning (Miconi et al., 2020). We used the inspiration from this process to design a fully-connected neural network layer with variable weights. Those weights could be modulated based on a variety of inputs, but we focus in this work on testing their performance on input reliability as a modulating signal. A restricted structure of modulating inputs and effects on the modulated layer reduces the likelihood of severe over-fitting and the complexity of the estimation problem. This allowed us to train the neural network using datasets that are loosely preprocessed with a high incidence of missing data while achieving high performance. We tested our model and compared it to state-of-the-art imputation and learning with missingness techniques on a variety of datasets. We tested on both small benchmark datasets but focused more on massive healthcare datasets (ACTFAST and UK Biobank) due to their relevance in real-world settings. Additionally, we tested the effect of complete feature removal on the robustness of the network to investigate the effectiveness of the model when transferred into a low-resource setting with a lower capacity of generating a complete set of readings. Overall, our model was more capable of producing accurate outputs despite signal degradation. It also showed more robustness as missingness levels were increased at test time and as we introduced out-of-distribution missingness patterns. In addition, the modulation network is

capable of handling data with variable quality metrics which makes it a go-to generic solution to different data quality issues.

## 2  Related work

The most obvious use case we propose for the modulated fully connected layer (MFCL) is handling missing data. There is a vast literature on imputation, which also attempts to use alternative inputs to replace missing data. Classical simple methods of imputation include constant values (e.g. mean imputation), hot deck, k-nearest neighbor, and others (Buck, 1960). Single or multiple imputation using chained equations (Gibbs sampling of missing data) is popular due to its relative accuracy and ability to account for imputation uncertainty (Azur et al., 2011). More advanced yet classic methods such as Bayesian ridge regression (MacKay, 1992) and random forest imputation (Stekhoven and Bühlmann, 2012) have seen relative success. Deep learning-based imputation that has been used recently uses generative networks (Beaulieu-Jones and Moore, 2017; McCoy et al., 2018; Lu et al., 2020; Lall and Robinson, 2021; McCoy et al., 2018; Mattei and Frellsen, 2019; Yoon et al., 2018; Ivanov et al., 2019) and graph networks (You et al., 2020). Our modulation approach has the potential to be incorporated into these existing deep-learning imputation methods to improve their performance and stability. Additionally, it provides the flexibility of skipping the imputation step altogether when the task performed does not require imputation (i.e. classification) thus skipping a preprocessing step and saving processing time.

Incorporating uncertainty measurements into deep neural networks has also been approached with Bayesian deep learning methods, (Wang and Yeung, 2016; Wilson, 2020) which has a complex, assumption-laden structure using probabilistic graphical models. One simpler variation of Bayesian deep learning is the Gaussian process deep neural network which assigns an uncertainty level at the output based on the missing data so that inputs with greater missingness lead to higher uncertainty (Bradshaw et al., 2017). Our method makes use of meaningful missingness patterns as opposed to treating it as a problem that leads to lower confidence in outputs. The approach to learning despite missingness was also tackled in the Neumiss Architecture (Morvan et al., 2020) but the latter lacks in that it can only incorporate missing flags and not quality measures. More recently, a very similar approach to the MFCL was utilized for classification of corrupted EEG signals (Banville et al., 2022) where they calculated the covariance matrix of the signal and used it to generate weights. This specific application shows the potential for success for our generalized approach since in our model the ideal transformation is learned based on the task. Overall, our model tackles several issues with data quality (missingness and variable quality) with one simple solution that can be readily integrated into existing models to improve their robustness.

## 3  Methods

### 3.1  Architecture

A fully connected layer has a transfer function of

$$h_{\text{out}} = f(\mathbf{W} \cdot h_{\text{in}} + b), \tag{1}$$

where $h_{\text{in}}$ is the input to the layer, $\mathbf{W}$ is the weight matrix, $\mathbf{b}$ the bias and $f$ the non-linearity function. $\mathbf{W}$ is optimized during training and fixed at inference. We propose a modulated fully connected layer (MFCL) where weights are made variable by replacing $\mathbf{W}$ by $\mathbf{W}_{\text{mod}}$ (Figure 1) where

$$\mathbf{W}_{\text{mod}} = g_W(m), \tag{2}$$

$$b_{\text{mod}} = g_b(m), \tag{3}$$

where $m$ is the modulating signal input and $g$ is the function that is defined by a multilayer perceptron. Combining equations 1, 2, and 3 and integrating the bias term into the weights, we get

$$h_{\text{out}} = f(g_W(m) \cdot h_{\text{in}}). \tag{4}$$

This form constitutes an interaction term between $m$ and $h_{\text{in}}$ which allows a multiplication operation that is not possible to compute by neural networks and can only be approximated. This additional functionality allows for a flexibility to incorporate interaction terms between an input and its quality measures in a more compact form.

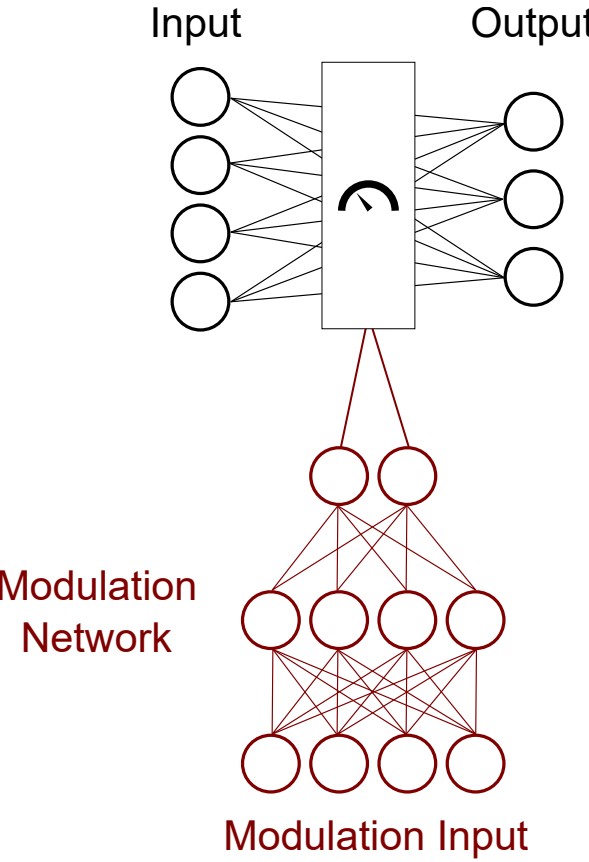

Figure 1: Schematic of modulated fully connected layer. The weights of the fully connected layers are modulated by the output of the modulation network.

### 3.2 Experiments

We first examined the modulating behavior of the MFCL using a toy model with simulated data and a two-input logistic regression model with modulation. We then assessed the performance of the MFCL layer in classification and imputation tasks using healthcare data from various sources. These experiments used modulating signals of missing value flags and input reliability values of noisy data. We can think of missing values as a special case included in reliability where *missing* implies completely unreliable measurement. For the sake of clarity, we test the cases of missing values and noisy values separately rather than combining them. For baseline comparison, we employed models with matching architectures while swapping the first fully-connected layer with a MFCL. Base architectures were guided by previous best-performing imputation models in the literature. Modulation network architectures were optimized using a restricted grid search. A complete description of the architectures is elaborated in the appendix.

### 3.3 Datasets

The motivating dataset for our experiments derives from operating room data from Barnes Jewish Hospital's Anesthesiology Control Tower project (ACTFAST) spanning 2012–2018. The Human Research Protection Office at Washington University in St Louis, USA approved this study and granted a waiver of informed consent. The dataset contains preoperative measurements of medical conditions, demographics, vital signs, and lab values of patients as well as postoperative outcomes that were used as labels for supervised learning including 30-day mortality, acute kidney injury, and heart attack. The ACTFAST dataset was used in previous studies for prediction of 30-day mortality (Fritz et al., 2020; 2019), acute kidney injury, and other complications (Cui et al., 2019; Abraham et al., 2021; Fritz et al., 2018). For predictors, we utilized a subset of the input features of preoperative vital signs and lab values (15 variables). Table 2 (Appendix) shows a list of variables used and the missing percentages. Table 7 (Appendix) shows the distribution of outcome values, which have a large imbalance between positive and negative samples. We also used the Wisconsin Breast Cancer (BC) dataset for classification of tumors from features extracted from a fine needle aspirate of breast mass image (Mangasarian et al., 1995). We only utilized the first ten variables quantifying the mean of each of the measures extracted from the images. We also utilized the OASIS dataset for dementia prediction that was open-sourced on Kaggle (Marcus et al., 2010). In the data quality experiments we utilized the BC dataset with simulated noise. We also used data from UK Biobank (Sudlow et al., 2015) which includes measurements of spirometry that is accompanied by a quality control measure that was used as modulating signal along with missing data flags for spirometry and other input variables to predict chronic obstructive pulmonary disease (COPD).

### 3.4 Simulated data

We created a simulated dataset with 1000 samples of two inputs and one classification output to demonstrate the operation of the MFCL. The two inputs were IID and the output function depended on the missingness pattern. It followed the following equation:

$$y = \begin{cases} 1 & \text{if } x_1 \text{ missing, } x_2 > 0.5 \\ \text{sign}(x_1 + x_2) & \text{otherwise} \end{cases} \tag{5}$$

We then added a missingness criteria for $x_1$ whenever $x_2 > 0.5$ and then additionally removed 5% of $x_2$ randomly. We trained a logistic regression model with MFCL (Figure 2A) with missingness of variables as the modulation input and then plotted the transfer function when there is no missingness (Figure 2B) and when each of the inputs is missing (Figure 2C & D).

### 3.5 Classification task

We ran five experiments for classification using the ACTFAST, BC, and OASIS datasets. We tested the MFCL in the place of fully-connected (FC) layers at the input level. For modulation input at the MFCL, we utilized the missingness flags concatenated with the input signal. Exact architecture and training parameters after hyperparameter tuning are presented in Table 3 and 4 of the Appendix.

#### 3.5.1 Baselines

The baseline classifiers were four MLPs with matching hidden layer structures that were fed imputed values using different algorithms, namely: chained regression with Bayesian ridge regularization (Scikit Learn Iterative Imputer), missForest algorithm (Stekhoven and Bühlmann, 2012), hot-deck imputation, and VAEAC (Ivanov et al., 2019). We also tested one graphical network model GRAPE (You et al., 2020). We also tested the concatenation of missingness flags with the input where missing values are imputed using mean and hot-deck imputation methods to compare the performance of MFCL with one of the simple masking models. We also tested the Neumiss (Morvan et al., 2020) which employed the same concept of learning with missingness. Since our experimental datasets are tabular in nature where tree-based models are known to perform better than DNNs (Grinsztajn et al., 2022), we also tested the XGBoost model (Chen and Guestrin,

2016) which handles missing data internally. The XGBoost model was designed with 100 trees with a max depth of 6 and a learning rate of 0.3.

**ACTFAST**   We built three classifiers to predict 30-day Mortality, Acute Kidney Injury (AKI), and Heart Attack from the preoperative input features. We used the datasets with the inherent missing data for training and then tested the trained models with additional missingness artificially introduced in both random and non-random fashions. Non-random missingness was introduced by two methods: removing the largest values and by removing all the data points of a certain input feature.

**Breast Cancer & OASIS**   The Breast Cancer dataset was fully observable dataset while the OASIS dataset had sparse missingness, so for the training with missingness, we had to introduce artificial missingness. For the classifier with missing flags as modulating signal, we introduced non-random missingness into the training dataset by removing the highest quartile of each variable. At the testing phase we evaluated each model with additional missingness similar to the ACTFAST classifiers.

### 3.6   Classification task with reliability measures

For the classifier with reliability signal, we tested it on the BC and COPD datasets. Reliability signals differ from modulating flag in that they are continuous rather than binary. The higher the number indicates the higher level of noise added (or the lower the reliability).

**Breast Cancer**   We utilized the fully-observable dataset but added Gaussian noise with zero mean and variable standard deviation (SD) where the SD values were sampled from a uniform distribution between 1 and 10 standard deviations of each variable. The higher end of SD values is very large, simulating a spectrum of noisy to essentially missing data.

**COPD**   We selected four variable from the UK Biobank datasets as input all of which contained organically missing values. These variables are spirometry forced vital capacity (FVC), C-reactive protein, and systolic and diastolic blood pressure. We used these to predict the occurrence of COPD as defined by the existence of the diagnosis ICD-10 code in the medical records. FVC values, in addition to missingness, contained a quality control variable assessing the quality of the measurement in 3 levels (best, medium, and worst). We integrated these levels with the missingness by encoding the three levels from best to worst as 0.0, 0.1, 0.2 and the missing flag as 1.0.

### 3.7   Imputation task

We ran two experiments for imputation by an auto-encoder using the ACTFAST dataset. We utilized the predictor features described earlier. We added the MFCLs in the place of FC layers at the inputs of an autoencoder imputation system. All parameters of training were similar to the baseline autoencoder described below. Exact architecture and training parameters after hyperparameter tuning are presented in Table 5 and 6 of Appendix.

#### 3.7.1   Baselines

The baseline autoencoder for imputation was trained by adding artificial missingness to the input values at random at a ratio of 25%. The loss function at the output layer calculated the mean squared error between the output values and the original values of the artificially removed values. The naturally missing data was included in the training dataset but not included in the loss function due to the absence of a known value to compare to. The models were tested by removing 10% of the test data either randomly (random removal) or by removing the highest 10% quantile (non-random removal).

### 3.8   Classification on fully observable training set

We further tested the MFCL DNN model training on a fully-observable dataset to investigate whether it can operate in the no-missing-data conditions as well. We also tested the trained model on introduced

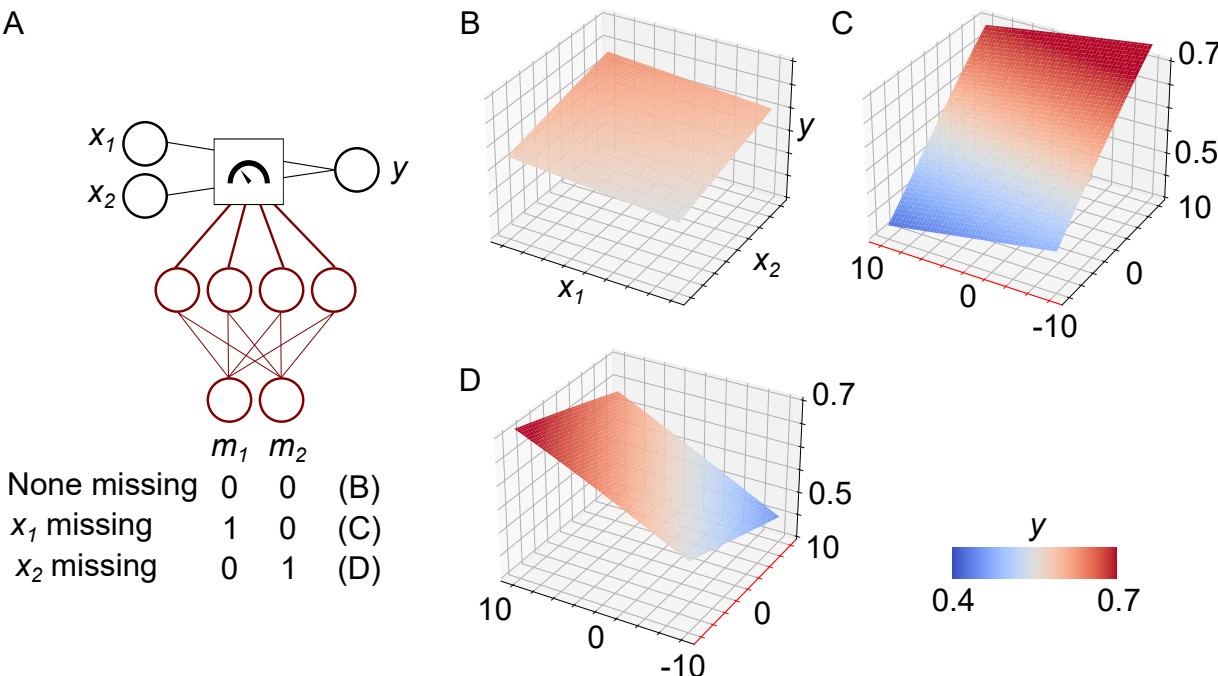

Figure 2: Simulation experiment results showing the operation of modulation. (A) The architecture of the modulation logistic regression model depicting the three test cases. (B) The input output relationship at the first test case when none of the inputs is missing. (C) The input output relationship when $x_1$ is missing, the missing input axis is labelled in red color and both the $z$-axis as well as the color map denote the output $y$. (D) The input output relationship when $x_2$ is missing.

missingness at the test stage only to test its robustness. Additionally, we trained another MFCL model with the fully observable dataset augmented with another copy with added 20% random missingness. We tested on the BC dataset as it was the only fully-observable one.

### 3.8.1 Baselines

We compared the two MFCL models with a DNN model with flags concatenated at the input and an XGBoost model that handles missing data internally. The parameters of the models and architecture were similar to those in the previous classification task.

### 3.9 Performance Evaluation

For the hyperparameter tuning of the modulating network, we divided the datasets into training, validation, and test sets with a split of 70:10:20. After the model was selected, we combined the training and validation sets into a training set (80:20 training test split) for each dataset to measure the performance of each of the architectures. We performed all our additional missingness tests only on the test split of the datasets. For classification tasks, we utilized area under receiver operating curve (AUROC) and area under precision and recall curve (AUPRC). In the training phase, binary cross-entropy loss was utilized as a cost function. For imputation tasks, we utilized mean squared error loss value as both the training cost function and the test performance evaluation metric.

To compute the margins of error, we conducted 1000 folds of paired bootstrapping for each experiment and computed the 95% confidence intervals for each test case. To test for statistical significance, we calculated

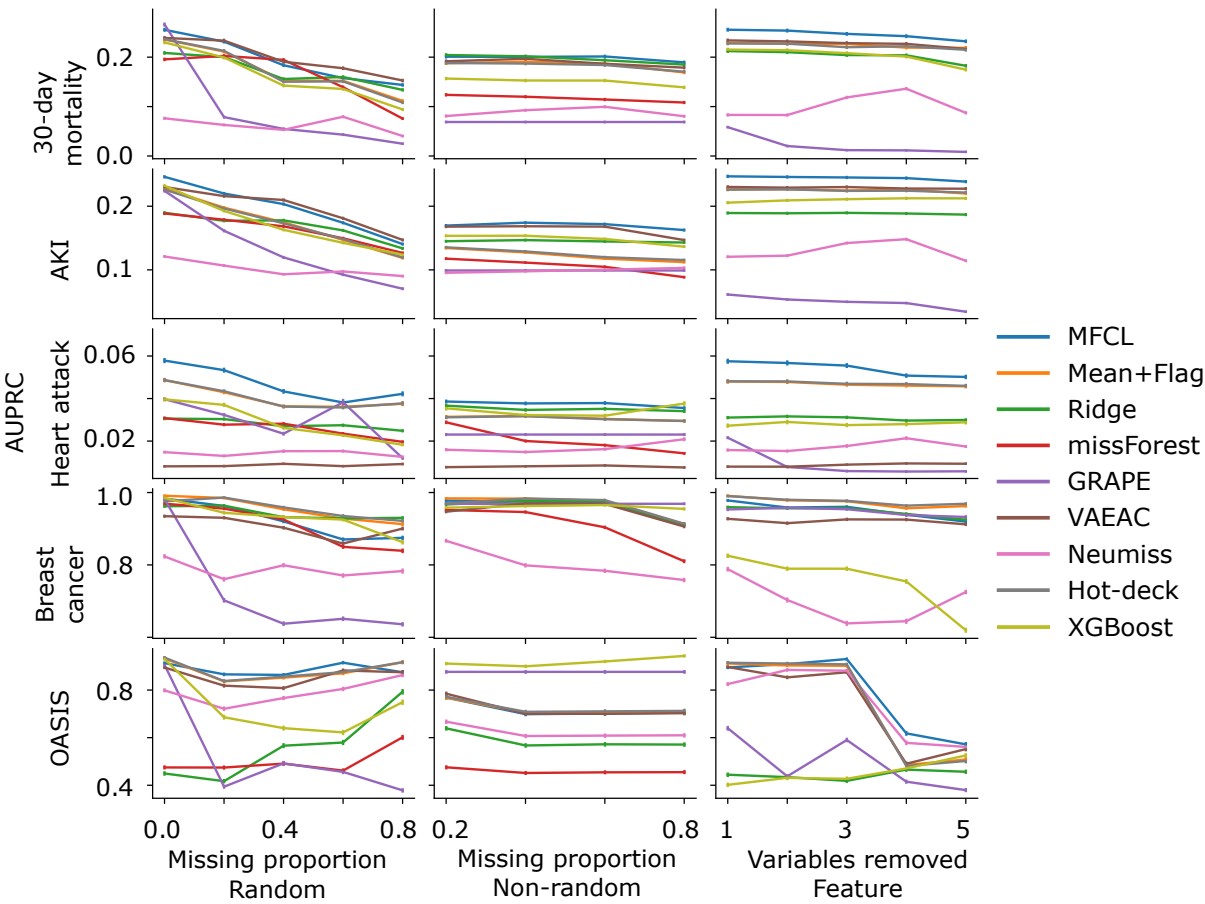

Figure 3: AUPRC of classification tasks with artificial introduction of random, non-random, and feature missingness averaged over all the bootstrap folds and test missingness levels (Error bars represent 95% confidence intervals).

repeated measure ANOVA on the bootstrapping results followed by paired t-test between different model pairs with correction for false discovery rate using Benjamini/Yekutieli method.

## 4 Results

### 4.1 Internal operation of MFCL

Figure 2 shows the results of the simulation experiment. When both inputs are available, the output appears to be positively correlated equally with both inputs (Figure 2B). When the input $x_1$ is missing, the output is strongly correlated with the value of $x_2$ reflecting the condition when missingness is meaningful. When the input $x_2$ is missing, the output depends on $x_1$ only but with a lower slope since the missingness of $x_2$ was randomly generated and not meaningful but still the output depends on the existing variable. The slope on the missing value axis is non-meaningful since the missing value is internally replaced with a zero (mean value of normalized data). These result show that the introduction of modulation alters the behavior of the fully connected layer to reflect the effect of missingness whether it is meaningful or random.

For a more realistic investigation of weights in a real experiment, we visualized the weights in one of the subsequent classification experiments (Figure 8). The weight patterns exhibit a different variation with each missing variable introduction with some variables being more important than the others. The model also

makes use of the mean-imputed missing variable. We can see that the signs of weights are mostly conserved indicating the redundancy of information supplied by the different measurements.

## 4.2 Classification with missing values

We then investigated whether modulation is beneficial to the classification performance in biomedical datasets with missing data. Figure 3 plots the mean test AUPRC of MFCL and baseline classifiers across all the testing conditions. Results for AUROC are shown in Figure 7. Results of the best-performing models are summarized in Table 1. We tested three paradigms of missingness with multiple levels of test missingness in each paradigm.

### 4.2.1 Random missingness

The first paradigm was random missingness where in the test phase, we removed 20%, 40%, 60%, and 80% of the data in addition to the existing missingness. We found that that MFCL provided the best AUPRC and AUROC in acute kidney injury (AKI), heart attack, and OASIS datasets. VAEAC imputation provided best performance in 30-day mortality (AUPRC and AUROC) and tied with MFCL at AKI AUPRC. Hot-deck imputation performed best in breast cancer dataset and tied with MFCL in OASIS AUROC measure.

### 4.2.2 Non-random missingness

The second testing paradigm, we tested a non-random missingness pattern where the highest quantile of each input was removed. We removed the highest 20%, 40%, 60%, and 80% of the values and calculated the mean AUPRC and AUROC. In this condition, MFCL had the best AUPRC performance across removals in the 30-day mortality, AKI, and heart attack datasets achieving also the highest AUROC in the latter two datasets. GRAPE imputation provided the highest performance in the breast cancer dataset (both AUPRC and AUROC) and OASIS (AUROC). VAEAC achieved the best AUROC in 30-days mortality and best AUPRC in OASIS dataset.

### 4.2.3 Feature missingness

The final testing paradigm was aimed at testing for complete missingness of certain features. This could happen when data from different sources are combined with variable data collection capabilities. We tested the removal of one to five features and calculated the AUPRC and AUROC of classification across different missingness levels. The missForest algorithm failed when a feature was completely removed so it yielded no data. MFCL models achieved the best AUPRC in all but the breast cancer dataset and the best AUROC in all but 30-day mortality and breast cancer. Addition of missingness flags along with mean imputation at the input achieved the best AUROC at the 30-day mortality dataset while hotdeck imputation achieved the best AUPRC and AUROC in the breast cancer dataset.

When MFCL model was not the best-performing model, it was still among the top-performing models indicating that even in the cases where it was not the highest performer, it did not fail completely. Other models had significant failures such as GRAPE that can be seen to fail at the feature removal. Other models such as VAEAC, hot-deck imputation, and addition of missingness flags had comparable performances but MFCL consistently was the top-performing model across the larger and more imbalanced datasets. MFCL, however, had poorer performance on the breast cancer dataset which was relatively smaller in size and the most balanced (Table 7). Another observation is that the model Neumiss which is similar in concept to MFCL showed the worst performance out of all the models which is quite surprising.

These results show that MFCL produces additional robustness against large quantities of non-random missingness while still performing strongly well where missingness is low, especially in precision which is most important in highly imbalanced datasets such as ACTFAST.

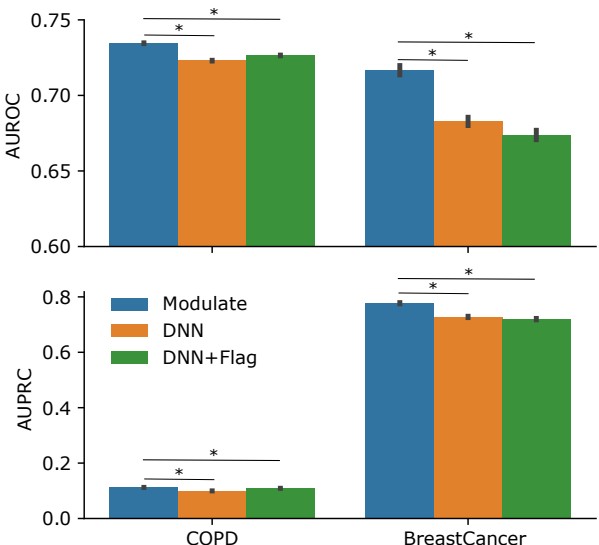

Figure 4: Performance of MFCL and baseline models (represented by AUROC and AUPRC) on the input reliability modulation tasks (Error bars represent 95% confidence intervals over bootstrapping folds). The asterisks indicate a statistically significant difference between the models marked by the ends of the line below the asterisk.

Table 1: Best-performing models across different missingness levels for each paradigm as tested by pairwise $t$-tests.

| MEASURE | DATASET | RANDOM | NON-RANDOM | FEATURE |
|---------|---------|--------|------------|---------|
| AUPRC | 30-DAY MORT. | VAEAC | MFCL | MFCL |
| | AKI | MFCL/VAEAC | MFCL | MFCL |
| | HEART ATTACK | MFCL | MFCL | MFCL |
| | BREAST CANCER | HOT DECK | GRAPE | HOT DECK |
| | OASIS | MFCL | XGBOOST | MFCL |
| AUROC | 30-DAY MORT. | VAEAC | VAEAC | MEAN |
| | AKI | MFCL | XGBOOST | MFCL |
| | HEART ATTACK | MFCL | MFCL | MFCL |
| | BREAST CANCER | HOT DECK | GRAPE/XGBOOST | HOT DECK |
| | OASIS | MFCL/HOT DECK | XGBOOST | MFCL |

### 4.3 Classification with input values with variable reliability

One of the novelties of this architecture is the possibility of embedding reliability measures in combination with missingness flags. We tested the modulation layer where input reliability (concatenated with the input signal) is used as a modulating signal instead of missing flags (Figure 4). We tested the breast cancer dataset with simulated noise added and with real-world reliability measurements in the COPD dataset along with missing data flags. We compared the model to a normal DNN model and another DNN with the data quality flags concatenated with the input. In this condition, the MFCL outperformed all other architectures over both AUROC and AUPRC measures significantly (Figure 4).

### 4.4 Autoencoder imputation with missing values

We tested imputation on the ACTFAST dataset (Figure 5) by introducing 10% missingness in the test datasets and measuring the mean squared error (Loss). We found that the addition of modulation layer did not add much to the imputation performance in comparison to the normal autoencoder for random removal. However, for the non-random removal, the MFCL-powered autoencoder showed significantly higher performance. It appears that all the networks were able to learn that representation indicated by the lower loss in the non-random removal case but with MFCL learning was superior. These results show that the addition of MFCL layer improved the imputation performance of autoencoders.

### 4.5 Classifier training with fully-observable data

We trained the models on a fully-observable (no missing data) dataset and tested on both fully-observable and introduced missingness. The MFCL model was very robust to training without change in the missing flags outperforming both the DNN with flags and XGBoost (Figure 6). Its performance was also robust with the introduction of artificial missingness in all the paradigms. Performance only had a sharp decline at high missingness ratios starting at 0.6 in the random missingness and 0.8 in the non-random missingness. Augmenting with missing data helped to decrease the rate of decline at those higher missingness ratios but overall MFCL outperformed the other models.

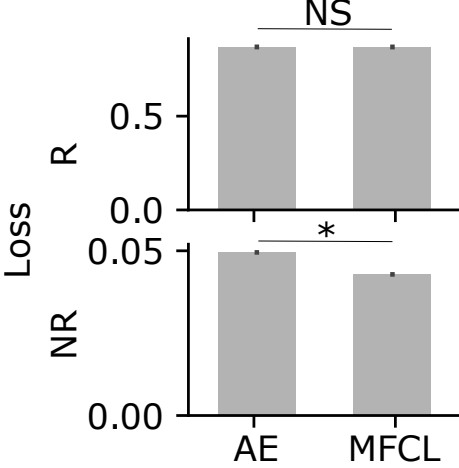

Figure 5: Performance (mean-squared loss) on imputation tasks with artificial introduction of missing data in random (R) and non-random (NR) fashions (Error bars represent 95% confidence intervals of bootstrapping folds). The asterisks indicate a statistically significant difference between the models marked by the ends of the line below the asterisk.

## 5 Discussion and conclusion

We propose a new layer for artificial neural networks inspired by biological neuromodulation mechanisms (Harris-Warrick and Marder, 1991). It allows the neural network to alter its weights and thus behavior based on the modulating signal (Figure 2). Our experiments showed that when added to standard architectures, modulating input layers make predictions that are more robust to missing and low quality data. Modulation was useful when missingness was introduced across different paradigms indicating the usefulness of modulation as a technique for introducing robustness into a system. This could provide an explanation of the existence of neuromodulation since it allows a compact and flexible implementation of the multiplication operation meaning it is possible to implement Bayesian inference in one layer. This was shown to be useful for multiple biological operations such as learning (Swinehart and Abbott, 2005),

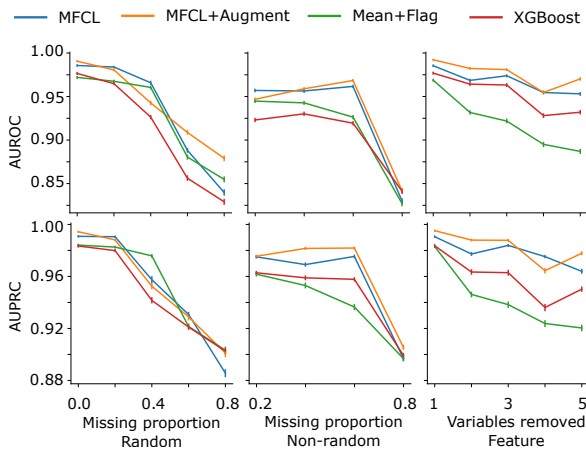

Figure 6: Performance of MFCL and baseline models (represented by AUROC and AUPRC) on the classification task using fully-observable training dataset and testing using the three missingness paradigms (Error bars represent 95% confidence intervals over bootstrapping folds).

locomotion control (Stroud et al., 2018), context modulation (Podlaski et al., 2020), and reinforcement learning (Miconi et al., 2020).

Modulation was shown to have a very strong performance, especially in non-random removal. This strong performance was more pronounced in the complete feature removal. This could be useful when data from different sources are combined such as deploying healthcare machine learning models to hospitals with limited resources. In lower resource settings, marginalized groups have been observed to have more missing data (Chen et al., 2020). Prediction methods not accounting for missing data can produce inaccurate results for these groups and hence, disadvantaging them. Therefore, methods that explicitly account for missing and low quality data instead of discarding the data are better in terms of social equity. On the other hand, non-transparency of neural networks, especially that use only small amount of data points for feature values can lead to feature-wise bias amplification (Leino et al., 2018). MFCL could improve the equity of machine learning systems and mitigate biases that arise due to socioeconomic differences between different communities. Another advantage over imputation techniques is that it also forgoes imputation operation leading to a decrease in computation power needed making the model suitable for low resource settings.

Despite our best effort, our testing procedure was limited by multiple factors discussed below. First, due to the novelty and flexibility of this model, there are many possible combinations for hyperparameters to explore. In order to limit the hyperparameter search space, we fixed the main network architecture and only varied the modulation network hyperparameters, but in practice there may be interactions between the hyperparameters of the two component networks. One other limitation is the lack of availability of large open tabular datasets with high missingness which limits the ability to generalize our findings. To make our experiments with informative missingness comparable across features, we restricted our input space to numeric variables and discarded categorical variables. Although our method could be applied to missing categorical variables, usually creating a "missing" level is fairly effective. Small technical modifications would also be required to modulate all features derived from encoding a categorical variable in the same way. We tested the application of the modulation process only in fully connected layers which are limited by nature in the types of data that they can handle. We intend to test the inclusion of modulation into other architectures such as convolutional layers and gated-recurrent units. It is important to address the issue of the high number of parameters in the modulation network. We did not search over regularization strategies of the modulation network, which could further improve its performance. Additionally, further

improvements to the layer could enhance this performance such as data augmentation or noise injection which is a subject of future work.

The main benefit of the modulation strategy compared to the conventional strategy of imputation is the high performance accompanied by a saving of a processing step. Its modular form also allows its addition to any architecture in a plug-and-play fashion. In future work, we also plan to test its introduction to the state-of-the-art imputation models and investigate their utility in data imputation, especially with its relative stability across different tasks in comparison to the other methods. The other main appeal to this method is its flexibility to the addition of reliability measures in combination with missingness flags.

One extension of our approach is to add the MFCL in locations in the network beyond the input layer. Preliminary experiments placing MFCL layers deep in the autoencoder experiments did not yield visible improvement (data not shown). The modulating signal could also be any input signal (not only reliability signal) such as context signal in a context switching task which could yield this network useful in multi-task reinforcement learning problems among many other applications (Jovanovich and Phillips, 2018). It can also be useful in compressing multi-task networks by compressing the multiple outputs into one with modulating input acting as a switch to change behavior of the network based on the task in question (Kendall et al., 2018; Chen et al., 2018; Li et al., 2020).

In conclusion, we have demonstrated that a modulation architecture could benefit in training neural networks in avenues where data quality is an issue. It can lead to the advancement of the field of MLOps where the current major concern is the integration of machine learning systems into production environments and thus fulfilling a big portion of the potential of artificial intelligence systems in advancing state-of-the-art technologies.

## 6 Code and data availability

ACTFAST data is not available due to clinical data privacy. COPD data was extracted under UK Biobank application 61530. Breast Cancer data is available through scikit-learn and preprocessed OASIS dataset is available through the Kaggle website `https://www.kaggle.com/datasets/jboysen/mri-and-alzheimers` while the raw data is available through the OASIS project `https://www.oasis-brains.org/`. Code is available under MIT license on Github: `https://github.com/mabdelhack/mfcl`.

## 7 Competing interest declaration

The authors declare no competing interests.

## 8 Acknowledgments

The authors would like to thank Alex Kronzer and Milos Milic for their assistance in data management and curation. The first author would like to thank Natalie Shreve for her tremendous help by offering him an internet connection as he moved countries in the middle of lockdown making the first version of this manuscript possible. This work was funded through the National Institute of Nursing Research (NINR) grant number 5R01NR017916-03. The UK Biobank data management was conducted under the auspices of UK Biobank application 61530, "Multimodal subtyping of mental illness across the adult lifespan through integration of multi-scale whole-person phenotypes."

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

## A  Appendix

The models were all built and tested using Pytorch 1.6 and run on a GeForce GTX 1080 GPU (Nvidia Corporation, United States). The base networks were designed based on knowledge of previous literature that utilized the datasets we used in this paper. For the MFCL layer architectures, we tested a small subset of modulation architectures on the ACTFAST 30-day mortality data. We then fixed the base architecture for other ACTFAST tasks. We decreased the size of the architecture on other datasets to avoid overfitting. To perform the hyperparameter search, we split the data into training, validation, and test sets using a 70:10:20 ratio. Tested architectures for the modulation layer are as follows:

- 1 hidden layer with 8 neurons

- 2 hidden layers with 8 neurons each

- 3 hidden layers with 8 neurons each

- 3 hidden layers with 8-4-8 neurons

- 3 hidden layers with 16 neurons each

- 4 hidden layers with 8 neurons each

- 5 hidden layers with 8 neurons each

We found little differences (non- significant) and selected the highest-performing architecture. We then combined the training and validation sets to generate a new training set that was used on the final model training. We tried to avoid a large number of hyperparameter tuning as we attempt to test the stability of the new architecture in less than optimal conditions. The final architecture and training details for classification tasks are presented in Table 3 and Table 4. The final architecture and training details for imputation task are presented in Table 5 and 6.

Table 2: Input variables and missing percentages in ACTFAST datasets.

| Input Variable | Missing Percentage | | |
|---|---|---|---|
| | 30-day Mortality N=67961 | AKI N=106870 | Heart Attack N=111888 |
| Systolic Blood Pressure | 58.5% | 57.3% | 57.5% |
| Diastolic Blood Pressure | 59.0% | 57.8% | 58.1% |
| Heart Rate | 1.2% | 1.3% | 1.3% |
| SpO$_2$ | 1.0% | 1.1% | 1.1% |
| Alanine Transaminase | 67.5% | 65.4% | 66.1% |
| Albumin | 67.3% | 65.1% | 65.8% |
| Alkaline Phosphatase | 67.5% | 65.4% | 66.1% |
| Creatinine | 22.4% | 23.8% | 26.1% |
| Glucose | 20.2% | 21.7% | 23.4% |
| Hematocrit | 20.4% | 22.4% | 24.1% |
| Partial Thromboplastin Time | 61.5% | 59.3% | 60.2% |
| Potassium | 22.0% | 23.3% | 25.0% |
| Sodium | 21.9% | 23.3% | 25.0% |
| Urea Nitrogen | 22.0% | 23.4% | 25.1% |
| White Blood Cells | 22.2% | 23.9% | 26.2% |

Table 3: Architectural details of the networks used in the classification with missing values tasks.

| Dataset/Task name | # hidden layers (hidden units per layer) | Hidden layer activation function | Output layer activation function | Modulation layer location | Modulation layer # hidden layers (hidden units per layer) |
|---|---|---|---|---|---|
| ACTFAST (30-day Mortality) | 2 layers (8-4) | ReLU | Sigmoid | Hidden layer 1 | 3 layers (8-8-8) |
| ACTFAST (AKI) | 2 layers (8-4) | ReLU | Sigmoid | Hidden layer 1 | 3 layers (16-16-16) |
| ACTFAST (Heart Attack) | 2 layers (8-4) | ReLU | Sigmoid | Hidden layer 1 | 3 layers (16-16-16) |
| Breast Cancer | 2 layers (4-2) | ReLU | Sigmoid | Hidden layer 1 | 2 layers (8-8) |
| OASIS | 2 layers (4-2) | ReLU | Sigmoid | Hidden layer 1 | 2 layers (4-4) |
| COPD | 2 layers (4-2) | ReLU | Sigmoid | Hidden layer 1 | 2 layers (8-4) |

Table 4: Training parameter details of the networks used in all the classification tasks. SGD denotes Stochastic Gradient Descent.

| Dataset | Batch size | # of epochs | Optimizer | Learning rate | Momentum |
|---|---|---|---|---|---|
| ACTFAST | 64 | 50 | SGD | 0.001 | 0.9 |
| Breast Cancer | 64 | 50 | SGD | 0.03 | 0.9 |
| OASIS | 64 | 1000 | SGD | 0.01 | 0.9 |
| COPD | 64 | 50 | SGD | 0.03 | 0.9 |

Table 5: Architectural details of the network used for imputation tasks (both datasets).

| # hidden layers (hidden units per layer) | Hidden layer Activation function | Output layer Activation function | Modulation layer location | Modulation layer # hidden layers (hidden units per layer) |
|---|---|---|---|---|
| 3 layers (10-5-10) | ReLU | Linear | Hidden layer 1 | 3 layers (8-8-8) |

Table 6: Training parameter details of the network used for imputation tasks.

| Batch size | # of epochs | Optimizer | Learning rate | $\beta_1, \beta_2$ |
|---|---|---|---|---|
| 64 | 30 | Adam | 0.01 | (0.9, 0.999) |

Table 7: Output variables imbalance and the number of samples in classification datasets.

| Output Variable | Positive Percentage | # of Samples |
|---|---|---|
| 30-day Mortality | 2.3% | 67,961 |
| Acute Kidney Injury | 6.1% | 106,870 |
| Heart Attack | 0.9% | 111,888 |
| Breast Cancer | 62.7% | 569 |
| OASIS | 44.8% | 373 |
| COPD | 4.0% | 502,476 |

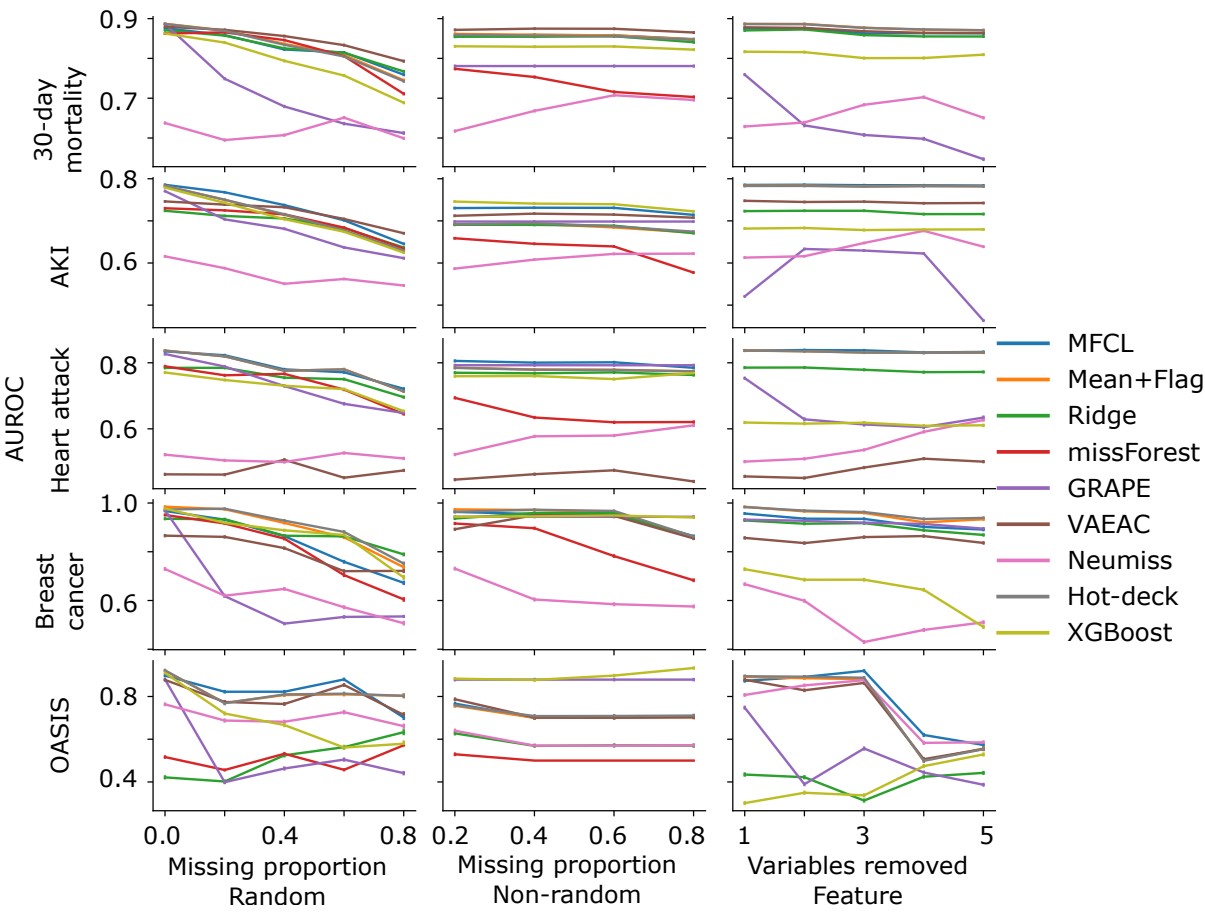

Figure 7: AUROC of classification tasks with artificial introduction of random, non-random, and feature missingness averaged over all the bootstrap folds and test missingness levels (Error bars represent 95% confidence intervals).

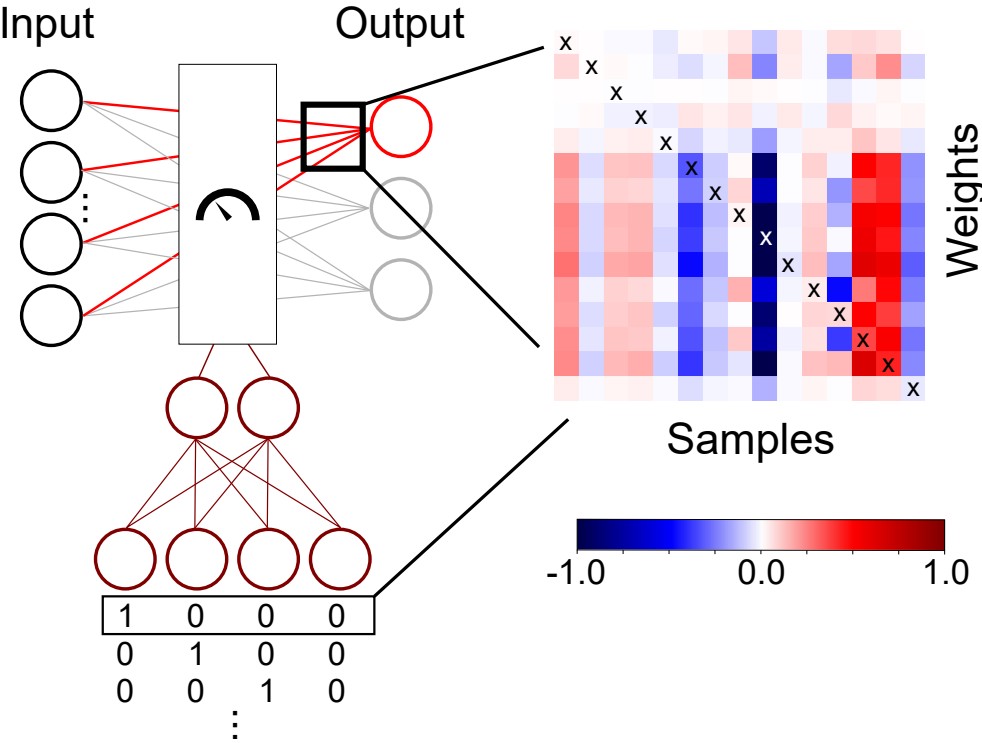

Figure 8: Visualization of variability of weights in the acute kidney injury MFCL classification model with the introduction of missing flags. The color map shows the percentage change of the weights in comparison to the no-missingness condition. Each row represents a condition where one of the fifteen inputs is missing (the missing variable is marked by the X mark).

