# OpenReview forum: "A Modulation Layer to Increase Neural Network Robustness Against Data Quality Issues"
_TMLR — Accepted by TMLR_

### Review · Reviewer_JsoF · 2023-01-22

**Summary Of Contributions:**

The work addresses the problem of learning from non-curated data, e.g. having missing/corrupted data.
To do this the proposed idea is to modulate the weights of a main network based on the output of a
data-driven function. Networks considered are fully connected. This process is motivated by neuroscience
where it was observed that the human brain can modulate the sensory input it receives depending on context.


**Audience:**

Yes

**Broader Impact Concerns:**

I am quiet concern about the broader impact of this work. The ML contribution is weak, and the experiments an relatively small datasets feel quiet ad-hoc. The experimental evidence of a solid ML contribution are therefore weak at this point.

**Claims And Evidence:**

Yes

**Requested Changes:**

Major
- Paper’s contribution is almost entirely experimental, which implies that experiments should be particularly rigorous and ambitious. In this respect, I have a number of concerns:
    - The experimental scenarios are relatively ad-hoc and do not simplify comparisons with other published works in the literature. I would have liked clear statements about how the results proposed here replicate or challenge previous experiments in other papers.
    - The range in which the parameters were searched is not given and it is not clearly written how the (validation?) data were used for model selection. How was the training loop stopped? Always 30 epochs? Any early stopping?
    - Main text suggests the batch size was always 64. If so, no need to repeat it everywhere in appendix. It also says that learning rate was set to 0.01 but I read 0.03 in the appendix. This makes me skeptical about the rigour of the experiments.
- Given the experimental nature of the work I would have expected some investigation on what the modulating function $g$ does, for example when missing data are introduced. This would be something like a sensitivity analysis.
- The method is evaluated a tabular data for which deep learning is still not considered the "go to" method by practitioners. Gradient boosted trees which natively support missing data during training should be compared to demonstrate the practical relevance of the work.
- The work is motivated by neuroscientific evidence of sensory modulations in the human brain. The input in such settings are images or sounds. To have experiments fully convincing for the broad ML community, results on such data are necessary.
- Error bars are mentioned in caption of figure 3 and 6 but they are not visible. Also I would expect ROC AUC scores to systematically decrease as more and more variables are removed (missingness is added) but the curves are not always decreasing which suggests a huge variance in the numbers reported. How much can we trust these line plots? Are the data too small to conclude?

Minor:
- Please consider using \mathrm for words in equations such as “out”, “in”, “mod”, “missing” or “sgn”. For example $h_\mathrm{out}$
- W mod is written with both mod in bold or not in eq 2 and above.
- “open-sources on Kaggle” -> “open-sourced on Kaggle”
- The architecture and training parameters could be more synthetically summarized in the table rather than in plain text in appendix

Reference:
I found this work https://www.sciencedirect.com/science/article/pii/S1053811922001239 that can potentially be related to this idea of weight modulation.


**Strengths And Weaknesses:**

The paper is relatively easy to follow and does a good job at positioning its contributions wrt the literature
in neuroscience and in machine learning.

Overall, the ML contribution is quiet limited. It consists of equation 4 and figure 1 where the weight matrix of a fully
connected network is multiplied element-wise by the output of a network fed by a “modulating” signal.

---

> ### Author Response · Authors · 2023-02-15
> **Summary of changes in revised manuscript**
>
> We would like to thank the reviewer for their thoughtful comments and actionable items. Below are point-by-point responses to the comments from the last review.
>
> > The paper is relatively easy to follow and does a good job at positioning its contributions wrt the literature in neuroscience and in machine learning. Overall, the ML contribution is quiet limited. It consists of equation 4 and figure 1 where the weight matrix of a fully connected network is multiplied element-wise by the output of a network fed by a “modulating” signal.
>
> Thank you very much for your accurate description. We believe that the simplicity of this solution is actually a strength point not a weakness. The architecture tackles a group of problems with data quality as well as ML equity using one simple solution that can be easily plugged into existing architectures. We also tested it on massive real-world datasets to ensure its effectiveness including showing where other models are outperforming. Of course, there will always be a lot of additional work that is to be desired but it is important at this point to put the work out for others to find new test cases some of which we allude to in the discussion.
>
> > Paper’s contribution is almost entirely experimental, which implies that experiments should be particularly rigorous and ambitious. In this respect, I have a number of concerns:
>
> Thank you for the comment and we apologize for giving the impression that the experiments were not rigorous. It is a product of adding and removing experiments in previous rounds of revision in previous conference submissions. We have revised the whole methods section to ensure consistency with appendices.
>
> > The experimental scenarios are relatively ad-hoc and do not simplify comparisons with other published works in the literature. I would have liked clear statements about how the results proposed here replicate or challenge previous experiments in other papers.
>
> Thank you very much for this comment. We have added statements in the introduction and methods sections to address this point. The reasoning behind the methods of comparison we utilized is to actually compare the practicality of use in real scenarios as opposed to the common practice in ML papers where well-curated benchmark datasets are used that end up not generalizing to real data. That is why we utilized ACTFAST dataset that was used to train a decision support system currently under clinical trial. In addition, we are presenting a system that not only deals with missing data but also handles data of low quality in one system which is a clear advantage over the existing imputation models. In that sense, the only model that is comparable is the DNN with modulation flags concatenated at input that we compare in the paper. Imputation and other models that explicitly deal with missing data are specific to handling missing data.
>
> > The range in which the parameters were searched is not given and it is not clearly written how the (validation?) data were used for model selection. How was the training loop stopped? Always 30 epochs? Any early stopping?
>
> We only ran the parameter search on the modulation layer and fixed the base architectures for fair comparison. The choice of hyperparameters was based on previous publications on the datasets so we used very similar parameters. We modified the methods section to make this clearer.
>
> > Main text suggests the batch size was always 64. If so, no need to repeat it everywhere in appendix. It also says that learning rate was set to 0.01 but I read 0.03 in the appendix. This makes me skeptical about the rigour of the experiments.
>
> Thank you for pointing out this confusing pattern. We had reported the training parameters for imputation task in the main paper but not for classification task. The example presented in the comment is from two different tasks. To avoid confusion we have moved training parameter details for imputation task also to the appendix and referred to the tables in the main text.
>
> > Given the experimental nature of the work I would have expected some investigation on what the modulating function does, for example when missing data are introduced. This would be something like a sensitivity analysis.
>
> Thank you very much for this comment. We added Figure 8 that shows the weight changes when missingness is introduced from one of the trained models (AKI classification).

---

> ### Author Response · Authors · 2023-02-15
> **Summary of changes in revised manuscript (Cont'd)**
>
> > The method is evaluated a tabular data for which deep learning is still not considered the "go to" method by practitioners. Gradient boosted trees which natively support missing data during training should be compared to demonstrate the practical relevance of the work.
>
> Thank you very much for this comment. We have added the results for the XGBoost model that natively supports missing data. The original motivation behind not adding such models in the original version was that the architectures would not be directly comparable to MFCL in a plug-and-play fashion as the DNNs but we clearly understand that such comparison would be made. We updated figure 3 to reflect the added XGBoost model.
>
> > The work is motivated by neuroscientific evidence of sensory modulations in the human brain. The input in such settings are images or sounds. To have experiments fully convincing for the broad ML community, results on such data are necessary.
>
> Thank you for this comment. Most of the neuroscientific knowledge of modulation emanates from the literature of invertebrates with a large portion of it in the form of chemical signaling which is very similar in nature to the biomarker data utilized here (except if we think of a freely moving animal where action and time would be a factor). Additionally, even in the case of human sensory modulation, it happens at the cue integration step which deals with high-level processing (the equivalent would be the last few layers of a multi-source DNN) which is not different from the current settings. For example, in the breast cancer datasets, the variables included are all aggregate measures from mammogram imaging. Processing images and sounds with missing regions in time or space employs vastly different processes of recurrent and top-down processing in the brain. It would actually be interesting to test the modulation on such tasks in which case we would need to make modulating convolutional/recurrent layers which would be a good future extension to test but out of scope for the current work.
>
> > Error bars are mentioned in caption of figure 3 and 6 but they are not visible. Also I would expect ROC AUC scores to systematically decrease as more and more variables are removed (missingness is added) but the curves are not always decreasing which suggests a huge variance in the numbers reported. How much can we trust these line plots? Are the data too small to conclude?
>
> Thank you for this comment. Actually increasing AUROC and AUPRC in uncurated data when removing data is not uncommon. There usually exists what is known as “hard” datapoints where inputs are counterintuitive to the output (contains outliers). In such a case, when data is removed and imputed, the imputation results do not contain those outliers thus increasing the performance. The reason the variance is small is due to how the bootstrapping is conducted where the training/testing was not repeated and the bootstrapped folds are drawn from the samples of that one testing instance. In such a case, the removal of samples from the “hard” samples would not change.
>
> > Please consider using \mathrm for words in equations such as “out”, “in”, “mod”, “missing” or “sgn”. For example ℎout
>
> > W mod is written with both mod in bold or not in eq 2 and above.
>
> > “open-sources on Kaggle” -> “open-sourced on Kaggle”
>
> > The architecture and training parameters could be more synthetically summarized in the table rather than in plain text in appendix
>
> Thank you very much for all these suggestions. We have implemented them in the revised manuscript.
>
> > Reference: I found this work https://www.sciencedirect.com/science/article/pii/S1053811922001239 that can potentially be related to this idea of weight modulation.
>
> Thank you very much for suggesting this work. We have included it in the related works section. It was published after the first version of this work was submitted for publication (and it did not cite our work) which is the reason why we were unaware of it.
>
> > I am quiet concern about the broader impact of this work. The ML contribution is weak, and the experiments an relatively small datasets feel quiet ad-hoc. The experimental evidence of a solid ML contribution are therefore weak at this point.
>
> Thank you very much for this comment. We would like to inquire about your suggestions to address this point. The datasets that we have used primarily (ACTFAST and UK-Biobank) are massive with over 100k samples each which is larger than the MIMIC datasets that are considered to be the gold standard for healthcare ML work. We added the number of samples in each dataset to table 7 to hone in on that point further. In addition, the ACTFAST dataset is one that was used to train one of the few ML-based decision support systems under clinical trials. We are also addressing the problem of equity and transfer learning to lower resource settings which is an understudied area in machine learning literature.

---

> > ### Comment · Reviewer_JsoF · 2023-03-03
> > **answer to authors**
> >
> > Dear authors,
> >
> > thanks a lot for clarifying a number of aspects. I would have still some remaining concerns that hopefully can be addressed:
> >
> > - I am not convinced by the argument that accuracy can go up with more missing data due to "hard examples". To me it suggests that not enough (train, val, test) random splits have been conducted to average out the effect.
> >
> > - thanks for adding XGBoost in your results but there is no mention of how parameters of XGBoost models have been set (number of trees, learning rate etc.). The code for XGBoost is also not present in the supplementary material.

---

> > > ### Author Response · Authors · 2023-03-04
> > > **RE: answer to authors**
> > >
> > > Dear reviewer,
> > >
> > > Thank you very much for the new comments. Below are our responses.
> > >
> > > > I am not convinced by the argument that accuracy can go up with more missing data due to "hard examples". To me it suggests that not enough (train, val, test) random splits have been conducted to average out the effect.
> > >
> > > Thanks for this comment. That is true, we did not run cross-validation in the test set. It was only conducted at the hyperparameter selection step. The testing was conducted on the holdout test dataset and the error bars are confidence intervals of bootstrapping on this set.
> > > We did not opt to do cross validation on this procedure as it would require nested cross validation which is computationally costly for the size of our datasets and is usually unnecessary as mentioned in this study: https://arxiv.org/pdf/1809.09446.pdf
> > >
> > > > thanks for adding XGBoost in your results but there is no mention of how parameters of XGBoost models have been set (number of trees, learning rate etc.). The code for XGBoost is also not present in the supplementary material.
> > >
> > > Thank you very much for your comment. We have added the parameters of XGBoost to the paper and are currenly cleaning up the code for adding it to the public code repo.

---

### Review · Reviewer_aHyJ · 2023-01-27

**Summary Of Contributions:**

The authors present a modification for fully-connected layers that replace fixed weights by a modulation function. Their modulated fully connected layer (MFCL) allows to modulate the weights for specific inputs depending on their reliability, e.g., missing values. Experiments show that the usage of MFCL can help to increase performance when missing values are present. Both settings were tested: 1. improve imputation model using MFCL, 2. do not use imputation but directly improve model using MFCL.

**Audience:**

Yes

**Broader Impact Concerns:**

None.

**Claims And Evidence:**

Yes

**Requested Changes:**

**Critical:**

- The authors use two methods for MNAR. Both seem very radical. There are others that could be more sensible. Could you elaborate why these make more sense than for example those used in:
	- https://www.semanticscholar.org/paper/A-Benchmark-for-Data-Imputation-Methods-J%C3%A4ger-Allhorn/98637661d9c5d9a6b749ed596b96e2d1fb1f9be3
	- https://www.semanticscholar.org/paper/JENGA-A-Framework-to-Study-the-Impact-of-Data-on-of-Schelter-Rukat/c6ca05d4805dbf0960427a0a01466c297606fe01
- Figure 5 shows the loss that "indicating higher performance". Directly show the performance

**For strengthening the work:**

- Section 4.2 lists the three paradigms. It would be helpful to use paragraph headings and name the results in Figure 3
- Figure 3 and Figure 4: It is not clear what the lines and asterisks mean
- instead of Figure 1, the authors could present how the modulation influences the weights. (Similar to Figure 2, which shows the final output y)

**Minor:**

- use capital letters for Figure x, Equation x, Table x, ...
- change “best performing“ -> “best-performing“, “top performing“ -> “top-performing“
- paragraph "COPD": missing "." at "blood pressure We used"

**Strengths And Weaknesses:**

**Strengths:**

- simple idea, easy to understand why this could help
- comparing usage of MFCL with and without imputation step
- good choice of baselines:
	- (with imputation) - different imputation approaches vs. MFCL
	- (without imputation) - DNN and DNN + missing flag vs. MFCL


**Weaknesses:**

- Some experiments could have been added to benchmark MFCL in more detail (There are papers indicating important differences)
	- What if datasets does not have any missing values? - Does it hurt to use MFCL
	- What if missing values only exist in test dataset? - Training on fully observed data
	- MCAR, MAR, MNAR - artificially corrupt (fully observed) datasets with theses could help to highlight pros and cons of MFCL

---

> ### Author Response · Authors · 2023-02-13
> **Summary of changes in revised manuscript**
>
> We would like to thank the reviewer for their thoughtful comments and actionable items. Below are point-by-point responses to the comments from the last review.
>
> > What if datasets does not have any missing values? - Does it hurt to use MFCL
>
> > What if missing values only exist in test dataset? - Training on fully observed data
>
> Thank you for these suggestions, we are adding the results of these experiment to the revised manuscript. We tested on the Breast Cancer dataset since it is the one that has a fully observed dataset it shows that the MFCL still works well. Adding missingness still resulted in decent performance though the addition of augmentation helped the performance at higher missingness ratios. We would be cautious at interpreting these results though given the small size of this dataset.
>
> > MCAR, MAR, MNAR - artificially corrupt (fully observed) datasets with theses could help to highlight pros and cons of MFCL
>
> Thank you for bringing this to our attention. Since the Breast cancer dataset originally didn’t have any missingness, we had corrupted the dataset for our experiments. We have updated the methods section in the main manuscript to clarify this point.
>
>
> >The authors use two methods for MNAR. Both seem very radical. There are others that could be more sensible. Could you elaborate why these make more sense than for example those used in:
> https://www.semanticscholar.org/paper/A-Benchmark-for-Data-Imputation-Methods-J%C3%A4ger-Allhorn/98637661d9c5d9a6b749ed596b96e2d1fb1f9be3
> https://www.semanticscholar.org/paper/JENGA-A-Framework-to-Study-the-Impact-of-Data-on-of-Schelter-Rukat/c6ca05d4805dbf0960427a0a01466c297606fe01
>
> Thank you very much for this comment. There were several reasons for using those methods:
>
> For feature removal, the motivation behind it is the equity in healthcare standpoint wrt available resources. We were interested to experiment with the idea that some healthcare providers will have access to fewer resources for measurement so we wanted to check how a model would transfer to such locations and thus the feature removal was tested.
>
> The MNAR method of removing data in a percentile range is the most common method of MNAR which is used in different parameters in the Jager and group’s paper (the first one). The high range of removals up to 80% of the data is quite common in the literature and generally serves as a stress test such as in:
>
> https://arxiv.org/abs/2010.16418
>
> https://openreview.net/pdf?id=J7b4BCtDm4
>
> And even in Jager and group’s paper, on one of the benchmarks that they utilized, the original paper used up to a 75% missingness ratio.
> We have amended the last part of the introduction to reflect this in the revised manuscript.
>
> > Figure 5 shows the loss that "indicating higher performance". Directly show the performance
>
> Thank you very much for your comment. We amended the language in the text to indicate that loss is the measurement of performance.
> For strengthening the work:
>
> > Section 4.2 lists the three paradigms. It would be helpful to use paragraph headings and name the results in Figure 3
>
> Thank you very much for this comment, we have restructured the paragraphs and figures.
>
> > Figure 3 and Figure 4: It is not clear what the lines and asterisks mean
>
> Thank you very much for this comment, we added descriptions of the lines and asterisks.
>
> > instead of Figure 1, the authors could present how the modulation influences the weights. (Similar to Figure 2, which shows the final output y)
>
> Thank you very much for this comment, we added a supplementary figure showing the actual change in weights in one of the trained networks.
>
> > use capital letters for Figure x, Equation x, Table x, ...
>
> > change “best performing“ -> “best-performing“, “top performing“ -> “top-performing“
>
> > paragraph "COPD": missing "." at "blood pressure We used"
>
> Thank you very much for these comments, the manuscript was updated to reflect these suggestions.

---

### Review · Reviewer_mYEV · 2023-02-13

**Summary Of Contributions:**

        This paper describes a neural network architecture to take into account extra training information related to the missingness values of certain inputs. For this a modulation network is proposed which outputs the main neural network weights to be used for prediction. The modulation network takes extra information as an input related to the missingness of certain attributes. The method is compared on several tasks with related methods from the literature that support handling missing values. The results obtained show that the proposed method performs better.


**Audience:**

Yes

**Claims And Evidence:**

Yes

**Requested Changes:**

        My main concern is that the potential benefit of the proposed architecture is not explored, vs. using a simple Neural Network which receives as an input concatenated to the data attribute the modulation information. I would suggest that the authors carry out some experiments, comparing with such a baseline, showing the benefit of the proposed architecture.


**Strengths And Weaknesses:**

Strengths:

        - Simple method proposed.

        - Extensive experimental comparisons.

Weaknesses:

        - Although mentioned, no error bars are given in the experiments. However, statistical tests to assess statistical significance are given.

        - It is not clear what is the benefit of using the architecture suggested by the authors vs. a NN which receives the modulation information concatenated to the input.

---

> ### Author Response · Authors · 2023-02-13
> **Summary of changes in revised manuscript**
>
> We would like to thank the reviewer for their thoughtful comments and actionable items. Below are point-by-point responses to the comments from the last review.
>
> > Although mentioned, no error bars are given in the experiments. However, statistical tests to assess statistical significance are given.
>
> Error bars are plotted in all figures. However, since they are very small they might not be very visible if the paper is zoomed out. We attempted to make them as clear as possible. We would appreciate any suggestions of actions we could take to make them more visible.
>
> > It is not clear what is the benefit of using the architecture suggested by the authors vs. a NN which receives the modulation information concatenated to the input. My main concern is that the potential benefit of the proposed architecture is not explored, vs. using a simple Neural Network which receives as an input concatenated to the data attribute the modulation information. I would suggest that the authors carry out some experiments, comparing with such a baseline, showing the benefit of the proposed architecture.
>
> This baseline is already included in all figures where it is labelled as "mean+flag" in the classification experiments and "DNN+flag" in the reliability modulation experiments. In addition, the hot-deck imputation model also has the missingness flag (modulation signal) concatenated with the input. We have modified the methods section (section 3.5.1) to make this point clearer.

---

> > ### Comment · Reviewer_mYEV · 2023-02-20
> > **Response to Authors**
> >
> > Thank you for your response. Given it, I do not have any concerns with the paper.

---

### Decision · Action_Editors · 2023-03-30

**Recommendation:** Accept as is

**Comment:**

Since the proposed approach was supported on the basis of experimental results, the main concern of the reviewers was the solidity and completeness of the experimental assessment, as well as its presentation. After discussion with the authors, all the issues raised by them have been resolved in a satisfactory way.

**Audience:**

The addressed problem is common in basically all real-world applications of neural networks, so it is of interest for many individuals in TMLR's audience.

**Claims And Evidence:**

The paper proposes a simple but effective way to cope with the problem of learning from data having missing or corrupted data, which is a very typical application scenario. Thus, the addressed problem is relevant and significant.
The proposed solution is based on a modulation function implemented by a  fully connected layer (MFCL) which modulates the weights for specific inputs depending on the presence or absence of values. The paper is well written and motivated, with a good covering of the related literature, including reference to relevant neuroscience literature.
Experimental assessment of the proposed approach is extensive and reports on the performance of the proposed approach in all relevant settings. With the help of reviewers, the experimental assessment has been further improved, which makes the paper ready for publication.